# Acute postoperative pain and dorsal root ganglia transcriptomic signatures following total knee arthroplasty (TKA) in rats: An experimental study

David E. Komatsu[1]*, Sardar M. Z. Uddin[1], Chris Gordon[2], Martha P. Kanjiya[2], Diane Bogdan[2], Justice Achonu[1], Adriana DiBua[2], Hira Iftikhar[2], Amanda Ackermann[2], Rohan J. Shah[2], Jason Shieh[3], Agnieszka B. Bialkowska[3], Martin Kaczocha[2,4]*

1 Department of Orthopaedics and Rehabilitation, Renaissance School of Medicine, Stony Brook University, Stony Brook, NY, United States of America, 2 Department of Anesthesiology, Renaissance School of Medicine, Stony Brook University, Stony Brook, NY, United States of America, 3 Department of Medicine, Renaissance School of Medicine, Stony Brook University, Stony Brook, NY, United States of America, 4 Stony Brook University Pain and Analgesia Research Center, Renaissance School of Medicine, Stony Brook University, Stony Brook, NY, United States of America

* David.Komatsu@stonybrookmedicine.edu (DEK); Martin.Kaczocha@stonybrook.edu (MK)

**Data Availability Statement:** All data are included within the manuscript and its Supporting Information files. All raw data files are available

## Abstract

Total knee arthroplasty (TKA) is the final treatment option for patients with advanced knee osteoarthritis (OA). Unfortunately, TKA surgery is accompanied by acute postoperative pain that is more severe than arthroplasty performed in other joints. Elucidating the molecular mechanisms specific to post-TKA pain necessitates an animal model that replicates clinical TKA procedures, induces acute postoperative pain, and leads to complete functional recovery. Here, we present a new preclinical TKA model in rats and report on functional and behavioral outcomes indicative of pain, analgesic efficacy, serum cytokine levels, and dorsal root ganglia (DRG) transcriptomes during the acute postoperative period. Following TKA, rats exhibited marked deficits in weight bearing that persisted for 28 days. Home cage locomotion, rearing, and gait were similarly impacted and recovered by day 14. Cytokine levels were elevated on postoperative days one and/or two. Treatment with morphine, ketorolac, or their combination improved weight bearing while gabapentin lacked efficacy. When TKA was performed in rats with OA, similar functional deficits and comparable recovery time courses were observed. Analysis of DRG transcriptomes revealed upregulation of transcripts linked to multiple molecular pathways including inflammation, MAPK signaling, and cytokine signaling and production. In summary, we developed a clinically relevant rat TKA model characterized by resolution of pain and functional recovery within five weeks and with pain-associated behavioral deficits that are partially alleviated by clinically administered analgesics, mirroring the postoperative experience of TKA patients.

from the Figshare database (accession number(s) 10.6084/m9.figshare.21288051) RNA-seq data have been deposited to the GEO database. The GEO Accession number for these data is GSE195833.

**Funding:** The study was funded in part by National Institute on Drug Abuse grant DA048002 and the Department of Anesthesiology, Renaissance School of Medicine, Stony Brook, NY, USA. Komatsu, Uddin, Gordon, DiBua, and Kaczocha received part of their salary from the National Institute on Drug Abuse grant DA048002. The funders had no role in study design, data collection and analysis, decision to publish, or preparation of the manuscript.

**Competing interests:** The authors have declared that no competing interests exist.

## Introduction

Pain arising from osteoarthritis of the knee is highly prevalent in the population and adversely affects both quality of life and worker productivity, costing the US economy over $150 billion annually [1–3]. Osteoarthritis (OA) is a progressive disease characterized by degradation of articular cartilage, subchondral bone damage, and inflammation of synovial tissues mediated by immune cell infiltration of joint tissues, extracellular matrix proteases, and pro-inflammatory cytokines [4]. OA is accompanied by significant pain that is commonly managed using topical nonsteroidal anti-inflammatory drugs (NSAIDs), in preference to oral NSAIDs, whose long-term utilization can result in adverse effects in multiple organs [5]. Total knee arthroplasty (TKA) is the current gold-standard intervention to alleviate pain and improve mobility in end-stage OA. TKA involves surgically replacing the articulating joint surfaces with prostheses. Approximately 700,000 TKAs are performed annually in the United States alone and this is projected to reach 1–2 million cases per year by 2030 [6].

TKA induces significant acute postoperative pain despite the use of multimodal analgesia [7–11]. Postoperative pain subsequent to surgical incision is functionally linked to increased sensitization and output of nociceptors, primary afferent neurons that transmit noxious stimuli from the site of injury to the spinal cord [12,13]. The development of new analgesics is hampered by our incomplete understanding of the molecular determinants of post-TKA pain, notably in dorsal root ganglia (DRG) that house the cell bodies of primary afferent neurons.

Elucidation of such pathways necessitates a preclinical TKA model that closely mimics the clinical procedure as well as outcomes including acute postoperative pain and recovery of function. While several TKA models have been reported in rats [14–17], none of these have demonstrated responsiveness to analgesia and complete restoration of knee function as reported in clinical studies. Herein, we describe a new preclinical model of TKA in rats that is accompanied by behaviors indicative of acute postoperative pain responsive to clinically employed analgesics, and exhibits functional recovery and pain resolution as observed in TKA patients. Demonstrating the utility of this model, mRNA sequencing (RNA-seq) of DRGs identified multiple transcripts and molecular pathways that are upregulated during the acute postoperative period, thus expanding our understanding of acute postsurgical pain.

## Materials and methods

### Ethics statement

All animal procedures were approved by the Stony Brook University Animal Care and Use Committee (#564663) prior to study initiation and met or exceeded United States Public Health Service Policy on Humane Care and Use of Laboratory Animals. Data reporting in the manuscript follows the recommendations in the ARRIVE guidelines.

### Prosthesis fabrication

To design the rat knee prosthesis, the tibial and femoral condyles of skeletally mature, 300 g Sprague Dawley rats were measured using digital calipers (Cen-tech) (Fig 1A and 1B). A tool and die corresponding to the curvature of the femoral and tibial implants were machined out of steel and heat-hardened. Implant outlines were then cut from flat, 1.5 mm thick 316 stainless steel. These pieces were positioned between the tool and die, loaded in a hydraulic press, and compressed to generate the prothesis curvature. A 1.5 mm diameter stainless steel stem was then spot welded to the bottom of each implant, and they were polished and autoclaved prior to implantation (Fig 1C). Each pair of implants weighed ~350 mg.

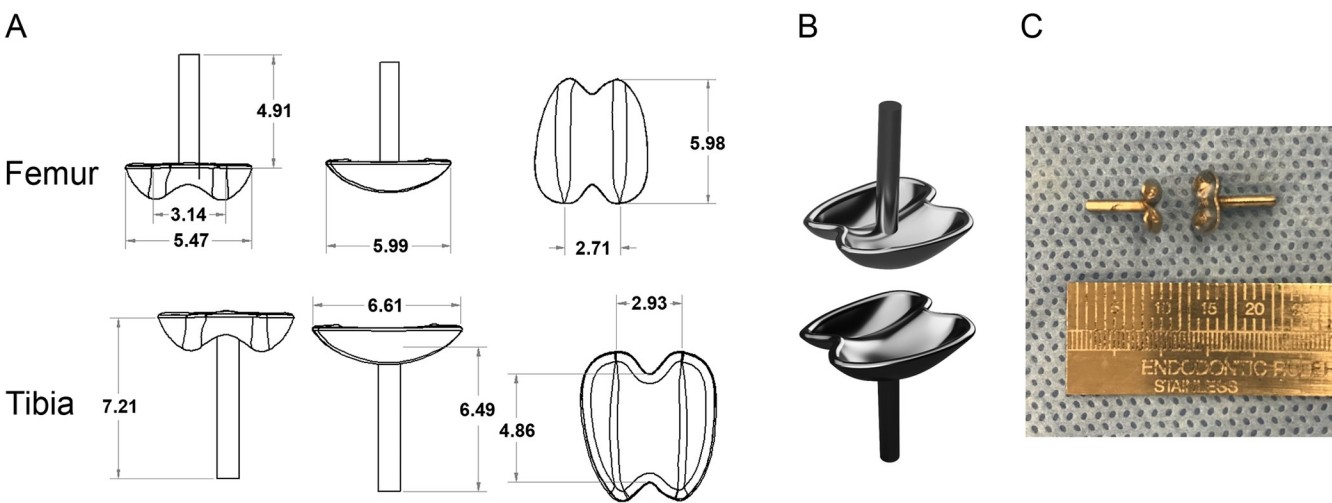

**Fig 1. TKA prostheses design.** Femoral and tibial prostheses were designed based on the dimensions of skeletally mature Sprague Dawley rats. (**A**) Measurements (mm) and schematic representation of the tibial and femoral prostheses. (**B**) Surface rendered model of femoral and tibial prostheses. (**C**) Photograph of the manufactured prostheses.

## Surgical procedures

Three-hundred-gram male and female Sprague Dawley rats (Envigo) were obtained and individually housed with *ad libitum* access to food and water. Lighting was maintained on a 12-hour light/dark cycle, temperature was kept at 22 ± 2°C, and humidity was constant at 50 ± 10% relative humidity. The animal cages were kept in the same location for the duration of each study. Rats were randomized to surgical groups by body weight and sex, and the behavioral outcome measures were performed in a random order each day by researchers who were blinded to the experimental conditions. All animals were monitored daily for one week after surgery and weekly thereafter for signs of distress and surgical site healing. At the end of the experiment, the animals were euthanized by $CO_2$ inhalation followed by decapitation.

## TKA surgery

For the TKA surgery, the rats were anesthetized with isoflurane and their left knees were sterilely prepped. An anterior midline incision (~20 mm) was made through the skin and fascia over the patella. A medial parapatellar incision was then made from the distal quadriceps to the tibial plateau and the patella was dislocated laterally to expose the knee (Fig 2A). The medial and lateral menisci were removed and the distal femur and proximal tibia were then reamed with a 1.3 mm k-wire (Zimmer) using a rotary cutting tool (Dremel, Robert Bosch Tool Corp) (Fig 2B and 2C). Next, the femoral and tibial condyles were trimmed using a diamond wafer blade attached to the rotary cutting tool (Fig 2D and 2E). Reaming and trimming were performed at slow rotation speeds under continuous sterile saline irrigation to remove debris and prevent thermal necrosis. The custom designed stainless steel femoral and tibial prostheses were press-fitted into the femur and tibia (Fig 2F). The patella was then reduced and the joint was closed using 4–0 resorbable sutures (Vicryl, Johnson and Johnson) (Fig 2G). The skin was then closed with 5–0 non-resorbable sutures (Prolene, Johnson and Johnson) (Fig 2H). Implant positioning was verified fluoroscopically (C-arm, Xi Tec) (Fig 2I and 2J).

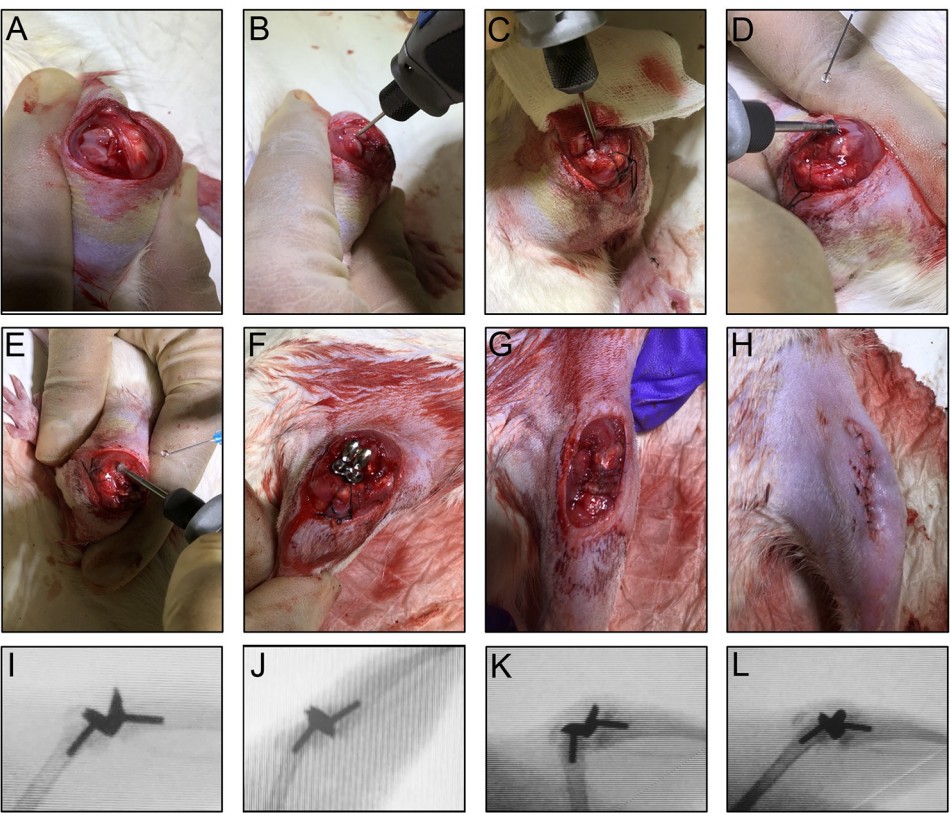

**Fig 2. Surgical procedure and implant positioning.** (**A**) Exposure of knee after anterior midline and medial parapatellar incisions. (**B**) Reaming of the femur. (**C**) Reaming of the tibia. (**D**) Trimming of the femoral condyle. (**E**) Trimming of the tibial condyle. (**F**) Knee after implantation of femoral and tibial prostheses. (**G**) Knee after patellar reduction and joint closure. (**H**) Knee after skin closure. (**I**) Fluoroscopic image of the knee in flexion immediately post-TKA. (**J**) Fluoroscopic image of the knee in extension immediately post-TKA. (**K**) Fluoroscopic image of the knee in flexion 5 weeks post-TKA. (**L**) Fluoroscopic image of the knee in extension 5 weeks post-TKA.

## Sham surgery

For Sham surgery, the rats were anesthetized with isoflurane and their left knees were sterilely prepped. An anterior midline incision (~20 mm) was made through the skin and fascia over the patella. The skin was then closed with 5–0 non-resorbable sutures. This procedure served as a control for both the TKA and DMM surgeries.

## DMM surgery

The destabilized medial meniscus model (DMM) [18,19] was used to induce OA. This is a surgical model of post-traumatic OA that replicates the clinical etiology of secondary OA and is not associated with the strong inflammatory response seen in chemically-induced OA models [20]. The rats were anesthetized with isoflurane and their left knees were sterilely prepped. An anterior midline incision (~20 mm) was made through the skin and fascia over the patella. A medial parapatellar incision was then made from the distal quadriceps to the tibial plateau and the patella was dislocated laterally to expose the knee. The medial meniscus was destabilized by transecting the anterior horn with a scalpel. The patella was then reduced and the joint was closed using 4–0 resorbable sutures. The skin was closed with 5–0 non-resorbable sutures and the animals received buprenorphine for postoperative analgesia. The rats underwent TKA surgery four weeks after the DMM or sham procedure as described above.

## Incapacitance

Rats were placed in a small animal incapacitance meter (IITC Life Sciences, US) and six discrete 15-sec recordings were collected on the indicated days as we previously reported [21]. The ratio of the weight placed on the left to the right hind limb is reported. The rats were acclimated to the apparatus for one week prior to TKA surgery and incapacitance was recorded at baseline and at the indicated time points.

## Home cage–locomotion and rearing

Locomotion and rearing activity during the 12-hour dark phase was quantified on the indicated days using the PAS Home Cage system (San Diego Instruments, US) as we previously described [21]. Rats were singly housed, and baseline, as well as post-TKA locomotion and rearing, were recorded for each animal. Normalized activity profiles (post-TKA/baseline) are presented for each group.

## Gait analyses

Gait analyses were conducted to assess the restoration of normal gait. An acrylic runway (120 cm long, 9 cm wide, and 30 cm high) was constructed to assess gait. A strip of white paper was cut to the length of the runway and placed at the bottom of the runway for each assessment. The hind paws of the rats were then dipped in nontoxic carbon ink (Platinum, Japan) and they were placed at the open end of the runway and allowed to walk to the darkened reward box on the other side. The paper strips were collected for calculation of SFI [22] and Stride Length.

## Sciatic Functional Index (SFI)

SFI was originally developed as a measure of sciatic nerve function and is based on measurements of toe spread and pawprint length [22]. We noted that rats limped following TKA surgery and this was reflected in pawprints that demonstrated reduced toe spread and shortened pawprint length. As such we used SFI as an index of gait recovery following surgery. To do so, the normal and experimental print length (NPL & EPL), toe spread (NTS & ETS), and intermediate toe spread (NITS & EITS) were measured. The right leg served as the control and the left as the experimental. Measurements were taken from three consecutive steps with a continuous gait. The mean of the three measurements for each parameter was then used to calculate SFI via the following equation [22]: SFI = -38.3 (EPL–NPL)/NPL + 109.5(ETS-NTS)/NTS + 13.3(EITS-NITS)/NITS– 8.8.

The rats were acclimated to this procedure for one week prior to surgery and SFI was calculated at baseline and at the indicated time points.

## Stride length

In addition to reduced toe spread and shortened pawprint length, we also observed that limping in rats was characterized by a reduced distance between steps. Therefore, we used stride length as an additional measure of gait recovery following surgery. Using the same strips, the distance between the front edge of the central paw pad was measured. The measurements were taken for the left and right paws from the same three consecutive steps used for SFI calculations. The mean stride length of the left paw was divided by the mean stride length of the right paw stride and the results reported as normalized stride length.

## ELISA

Whole blood samples were collected from the tail vein on the indicated days into BD Microtainer Tubes (SST Clear/Amber 365967) and serum separation was performed according to the manufacturer's instructions. Serum levels of TNFα (R&D Systems #RTA00) and IL-6 (R&D Systems #R6000B) were determined by ELISA. The plates were read on a microplate reader (SpectraMax i3X, Molecular Devices) and absorbance was read at 450 nm with wavelength correction set at 540 nm (SoftMax Pro 6.5, Molecular Devices).

## Drug administration

Morphine (Hikma), ketorolac (Cayman Chemical), and gabapentin (Acros Organics) were administered in a volume of 1 μl/g body weight. Morphine (1 mg/kg, dissolved in sterile saline) was given as a subcutaneous injection while ketorolac (10 mg/kg, dissolved in sterile saline) and gabapentin (100 mg/kg, dissolved in sterile saline) were given as intraperitoneal injections. Outcomes were measured prior to drug administration and one hour after administration.

## RNA sequencing

mRNA sequencing (RNA-seq) was used to explore gene expression profiles of L3-L4 DRGs collected from three male and female naive, sham, and TKA rats. Rats were euthanized by exsanguination after deep isoflurane anesthesia. Laminectomy was performed to allow direct access to the ipsilateral L3-4 DRGs. The DRGs were immediately preserved and stabilized in RNAlater (#R0901, Sigma). Total RNA was extracted using the PureLink RNA Mini Kit (#12183018A, Invitrogen/Thermo Fisher Scientific) following the manufacturer's instructions, and treated on column to degrade contaminating DNA with PureLink DNase (#12185–010, Invitrogen/Thermo Fisher Scientific). Purified total RNA was eluted from the column with RNase-free water. RNA integrity number (RIN) was measured using a BioAnalyzer (Agilent 2100) and all samples delivered a RIN value > 6.8. cDNA library construction and paired-end 150 bp sequencing were performed on Illumina NovaSeq platforms by Novogene (Davis Lab, Sacramento, CA) and subsequently tested for quality (Library QC) with Qubit, real-time PCR for quantification, and BioAnalyzer for size distribution detection. Pair-end reads were aligned to the rat genome (RGSC 6.0/rn6) using HISAT2, transcript counts were quantified with FeatureCounts, and differentially expressed genes (DEGs) were identified with DESeq2 package (Usegalaxy.org) [23]. Gene expression fold-change $\geq$ 1.5 and false discovery rate (FDR) < 0.05 were set as the threshold values for subsequent analysis. Gene set enrichment analysis with ClusterProfiler was performed in R (https://learn.gencore.bio.nyu.edu). Heatmaps were generated using Seqmonk (https://www.bioinformatics.babraham.ac.uk/projects/seqmonk). The GEO accession number for these RNA-seq data is: GSE195833

## Statistical analysis

A power calculation based on preliminary incapacitance data was conducted to determine the appropriate sample size. It was assumed that the resulting data would be normally distributed and One-way ANOVA would be used to assess differences of >10% between time points at alpha of 0.05 and power of 0.80. The power analysis was performed using an automated calculator [24] and revealed that a sample size of 8 would be required. Results are reported as group mean +/- standard error of the mean. Analgesic data were also transformed to % Reversal = (100*[After-Before]/[1-Before]) and reported as dot plots showing individual data points and group means (bars). Normality was first assessed using Shapiro-Wilk tests. Subsequently, comparisons between groups for longitudinal outcomes were made using One-way ANOVA

followed by Dunnett's multiple comparisons with the baseline data as the control group. Before and after comparisons for the analgesic responses were made using paired t-tests. All analyses were conducted on non-normalized data using Prism (Ver. 9, GraphPad) with significance set at $p < 0.05$.

## Results

### Prosthesis design and TKA surgery

The prostheses were designed based on the tibial and femoral condyles of skeletally mature Sprague Dawley rats and machined from stainless steel (Fig 1). TKA surgeries were successfully conducted as evidenced by full passive range of motion that was achieved postoperatively (Fig 2 and S1 Video). Moreover, the implants retained their position for the duration of the study (Fig 2K and 2L). Compared to baseline, the rats showed pronounced limping after TKA surgery (S2 and S3 Videos), which fully recovered by day 35 (S4 Video).

### Pain and functional recovery after TKA

We employed multiple approaches to assess postoperative behaviors indicative of pain and functional recovery including incapacitance, home cage locomotion and rearing, sciatic functional index (SFI), and stride length. In addition, we assessed systemic inflammation by quantifying serum levels of interleukin-6 (IL-6) and tumor necrosis factor alpha (TNFα), which are known to be elevated after TKA surgery in humans [25–28].

As expected, TKA induced pronounced incapacitance in the surgical limb, which did not recover to baseline levels until postoperative day 35 (Fig 3A). In comparison, sham surgery did not affect incapacitance (Fig 3A). In addition to weight-bearing deficits, animals experiencing

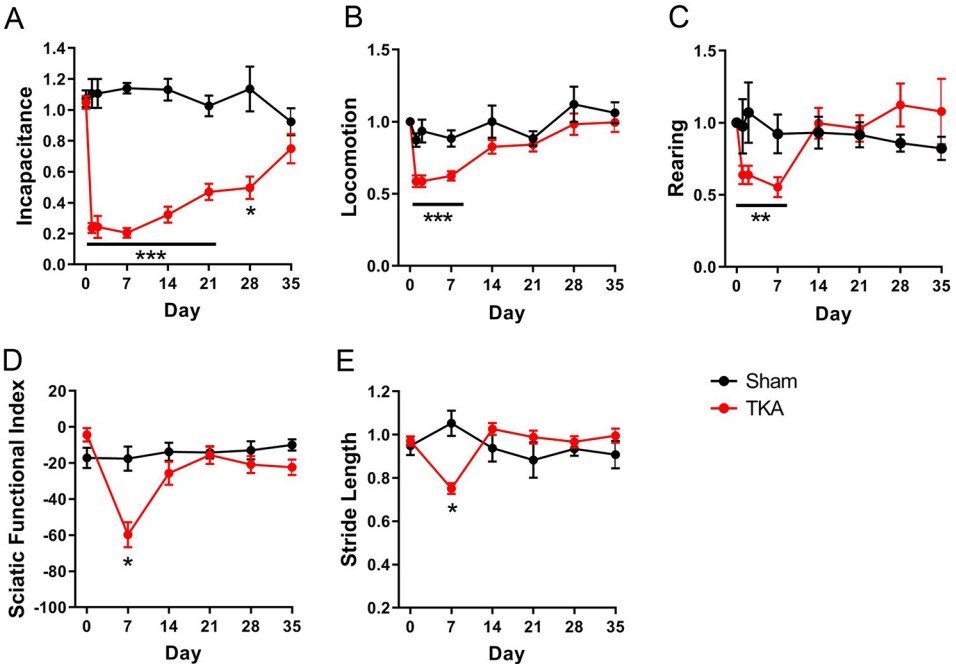

**Fig 3. Functional recovery after TKA surgery.** Recovery from surgery was evaluated over 35 days using a range of behavioral outcomes to compare rats subjected to TKA and sham surgery. (**A**) Incapacitance, (**B**) home cage locomotion, (**C**) rearing, (**D**) SFI, and (**E**) stride length. *, $p < 0.05$; **, $p < 0.01$; ***, $p < 0.001$ vs baseline, as determined by One-way ANOVA followed by Dunnett's post-hoc test (n = 8).

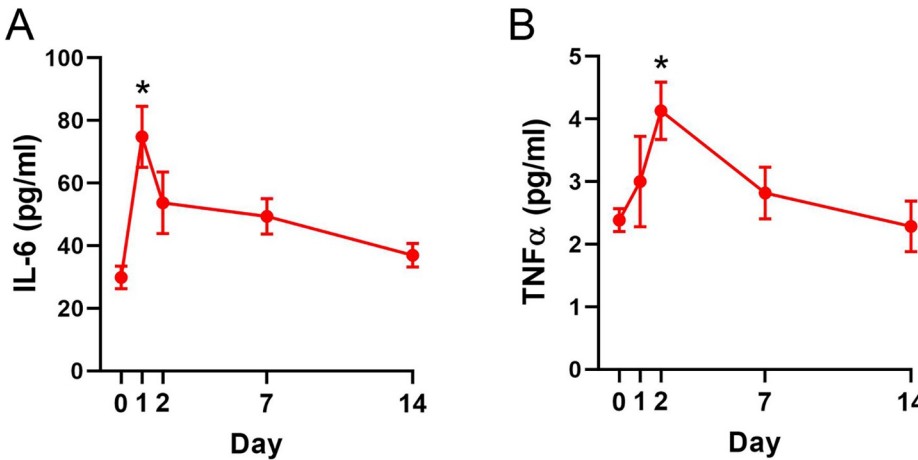

**Fig 4. Serum IL-6 and TNFα levels after TKA.** Levels of (**A**) IL-6 and (**B**) TNFα at baseline (day 0) and up to 14 days after TKA. *, $p < 0.05$ vs baseline as determined by One-way ANOVA followed by Dunnett's post-hoc test (n = 6).

postoperative pain exhibit reduced ambulation and rearing [21,29–32]. Consistent with these observations, home cage locomotion and rearing were diminished after TKA, with significant reductions in both of these outcomes for the first 7 days following surgery (Fig 3B and 3C). Again, no changes in either outcome were observed in the sham controls. Due to the pronounced limping seen in the first postoperative week (S3 Video), gait assessments began on day 7. Consistent with the home cage data, TKA rats exhibited significant reductions in SFI and stride length on day 7 that returned to baseline by day 14 (Fig 3D and 3E). No gait changes were observed in the sham group (Fig 3D and 3E).

TKA induces an acute inflammatory response as evidenced by elevated serum pro-inflammatory cytokines after surgery [25–28,33]. In agreement with the clinical data, serum cytokine analysis revealed elevated levels of IL-6 and TNFα after TKA surgery (Fig 4). IL-6 levels were significantly elevated on day 1 and then gradually returned to baseline (Fig 4). TNFα peaked on day 2 and subsequently returned to baseline.

## Acute postoperative pain and analgesic efficacy

As our TKA model induces functional deficits, we sought to determine if these deficits reflect postoperative pain. To test this, we administered morphine and ketorolac, two analgesics commonly used to manage acute postoperative pain. Incapacitance was assessed before and after drug administration on day 1, to reflect the immediate postoperative period characterized by elevated cytokine levels and on day 7, a time point at which inflammatory cytokine levels were no longer different from presurgical baselines (Fig 4). Administration of morphine or ketorolac individually partially restored weight bearing on day 1 with the combined administration of both analgesics showing increased efficacy (Fig 5A and 5B), consistent with clinical utilization of multimodal analgesia post-TKA. On day 7, morphine continued to exhibit efficacy in improving incapacitance while ketorolac did not, presumably reflecting reduced inflammation at this time point (Fig 5C and 5D). Given the lack of efficacy for ketorolac on day 7, the combination of morphine and ketorolac was not tested. Administration of vehicle had no effect on incapacitance.

The effects of morphine and ketorolac on gait were evaluated on day 7 due to the profound limping observed during the first week after surgery (S3 Video). Consistent with the incapacitance results, morphine improved SFI (Fig 5E and 5F), indicating that the deficits in SFI reflect

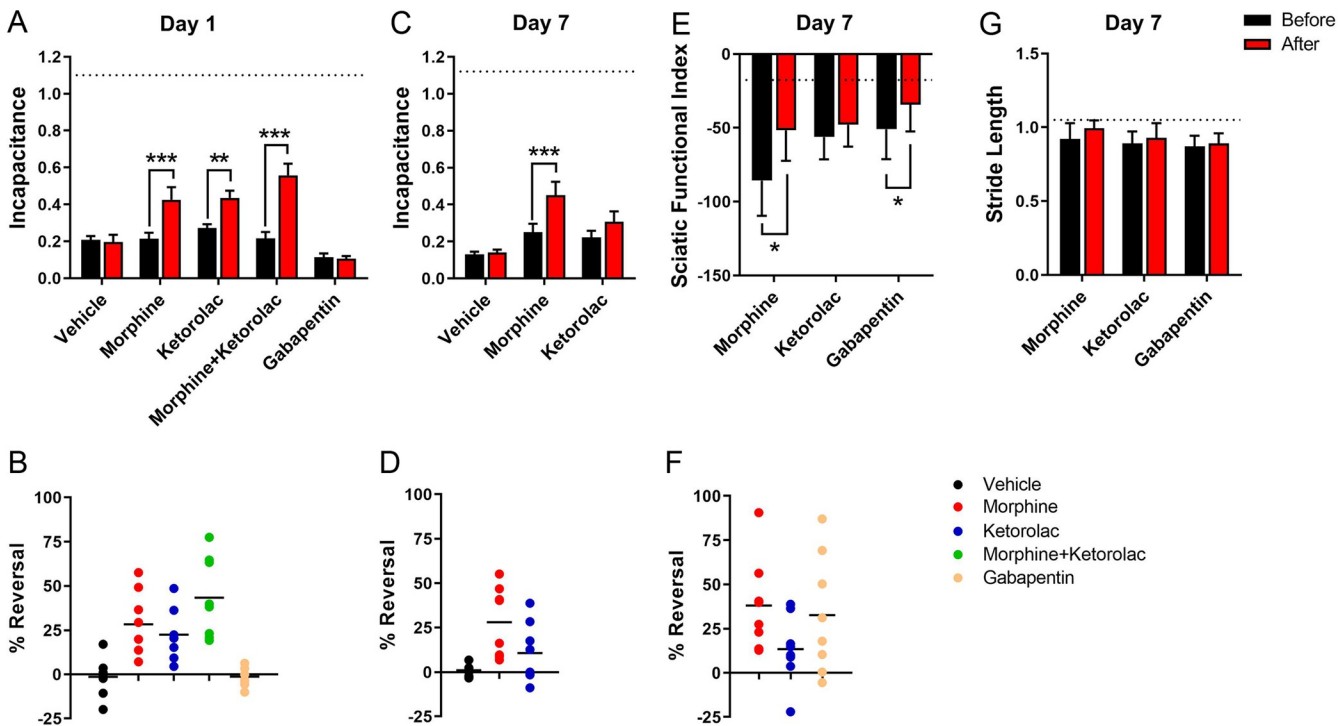

**Fig 5. Analgesic efficacy after TKA surgery.** (**A**) Incapacitance on day 1 before and after vehicle (saline), morphine (1 mg/kg), ketorolac (10 mg/kg), combined ketorolac and morphine, or gabapentin (100 mg/kg). Dashed line indicates the mean incapacitance value of the Sham group. (**B**) Percent reversal of incapacitance for each rat in **A**. (**C**) Incapacitance on day 7 before and after vehicle, morphine, or ketorolac. Dashed line indicates the mean incapacitance value of the Sham group. (**D**) Percent reversal of incapacitance for each rat in **C**. (**E**) SFI on day 7 before and after morphine, ketorolac, or gabapentin. Dashed line indicates the mean SFI value of the Sham group. (**F**) Percent reversal of SFI for each rat in **E**. (**G**) Stride length on day 7 before and after morphine, ketorolac, or gabapentin. Dashed line indicates the mean stride length value of the Sham group. *, $p < 0.05$; **, $p < 0.01$; ***, $p < 0.001$, as determined by paired t-tests (n = 8).

postoperative pain. Similar to incapacitance, ketorolac was also unable to rescue SFI on day 7. Stride length was unaffected by morphine or ketorolac (Fig 5G), indicating that this outcome measure has poorer sensitivity than incapacitance or SFI in measuring pain.

Next, we determined whether our model can discriminate between clinically effective and ineffective analgesics. Gabapentin is an anticonvulsant that is routinely added to multimodal analgesia protocols to manage acute postoperative pain after TKA, although recent meta-analyses indicate a lack of efficacy [34,35]. Consistent with these clinical observations, gabapentin failed to alter incapacitance (Fig 5A and 5B). Interestingly, gabapentin improved SFI, suggesting that there may be a neuropathic component to post-TKA pain in our model (Fig 5E and 5F). Collectively, these results demonstrate that our rat TKA model induces pain that can be partially alleviated by clinically efficacious analgesics.

## Influence of OA on pain and functional recovery after TKA

As OA is the major indication for TKA surgery, we assessed whether pre-existing OA influences recovery in our TKA model. OA was induced using the destabilized medial meniscus model (DMM) [18,19]. Four weeks later, the establishment of OA was confirmed by radiographic evidence of joint narrowing, histological evidence of cartilage degradation, and weight bearing deficits (Fig 6A–6C). Consequently, this time point was selected as the starting point for TKA surgery. Similar to TKAs performed in healthy rats, marked reductions in post-TKA incapacitance were observed in OA rats that persisted until day 21, with full recovery at day 35

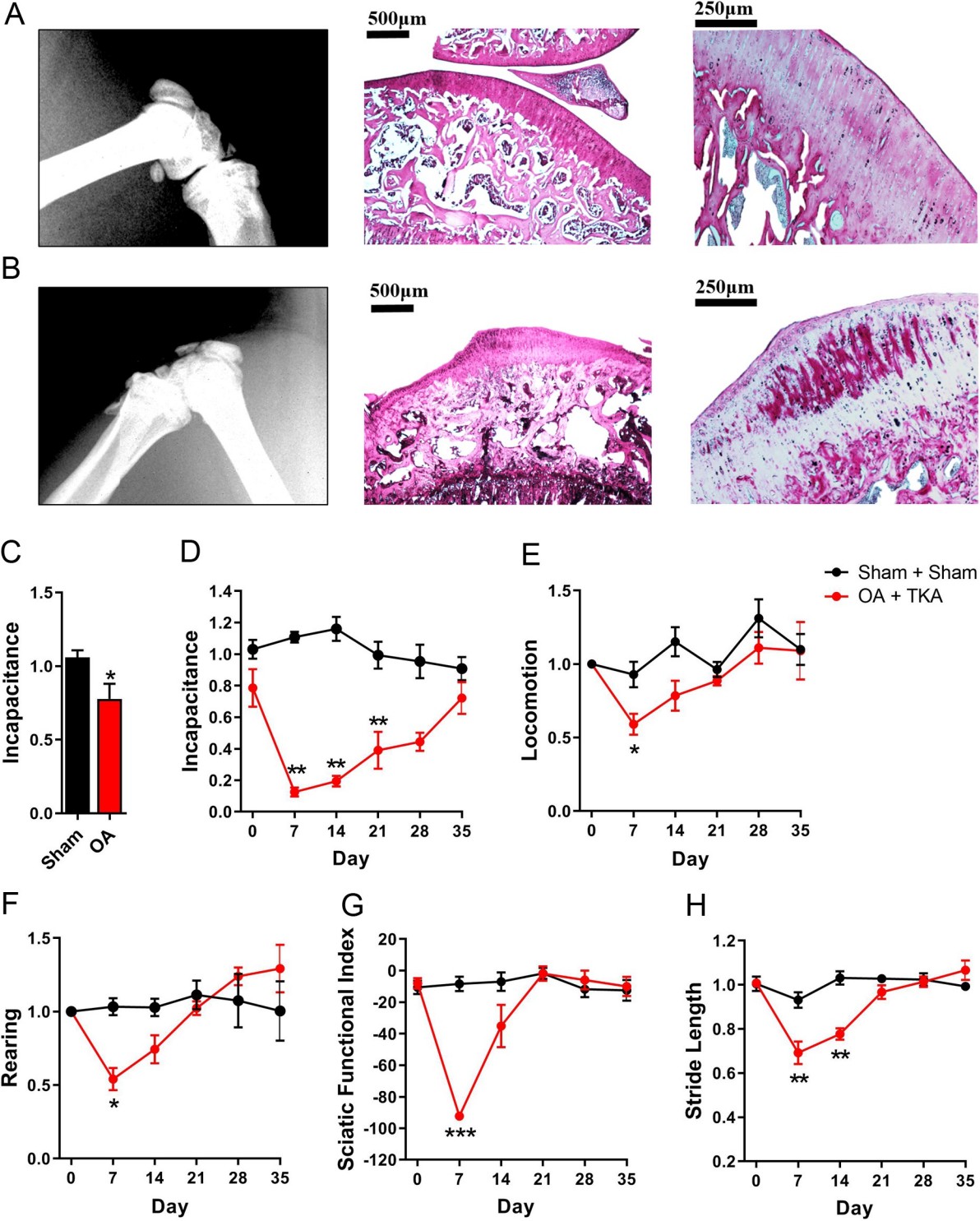

**Fig 6. Functional recovery after TKA surgery in rats with OA.** (**A**) Lateral knee radiograph (left), H&E (middle), and Sirius red (right) staining of the proximal tibia from a healthy rat. (**B**) Lateral knee radiograph (left), H&E (middle), and Sirius red (right) staining of the proximal tibia from an OA rat 4 weeks post-DMM surgery. (**C**) Incapacitance at 4 weeks after DMM or Sham surgery. $^*$, p < 0.05, as determined by unpaired t-test (n = 8). For **D-H**, rats underwent TKA or Sham surgery 4 weeks after OA induction via DMM (OA + TKA) or Sham (Sham + Sham) procedures, respectively. (**D**) Incapacitance, (**E**) home cage locomotion, (**F**) rearing, (**G**) SFI, and (**H**) stride length. $^*$, p < 0.05; $^{**}$, p < 0.01; $^{***}$, p < 0.001 vs baseline, as determined by One-way ANOVA followed by Dunnett's post-hoc test (n = 8).

(Fig 6D). Home cage locomotion and rearing were reduced at day 7 and recovered by day 14 (Fig 6E and 6F). Gait also demonstrated comparable recovery kinetics to healthy rats, with SFI significantly impaired on day 7 and stride length significantly impaired on days 7 and 14 (Fig 6G and 6H). None of these parameters were altered in the sham group. Collectively, our data suggest similar recovery patterns in rats with established OA compared to healthy rats.

## DRG transcriptomic signatures after TKA

To identify transcriptomic changes accompanying acute postoperative pain, we performed RNA-seq on DRGs obtained from naïve, sham, and TKA rats 24h after surgery. Hierarchically clustered heat map analysis revealed differences in the global transcriptomic landscape between the TKA and sham groups (Fig 7A). To identify differentially regulated transcripts,

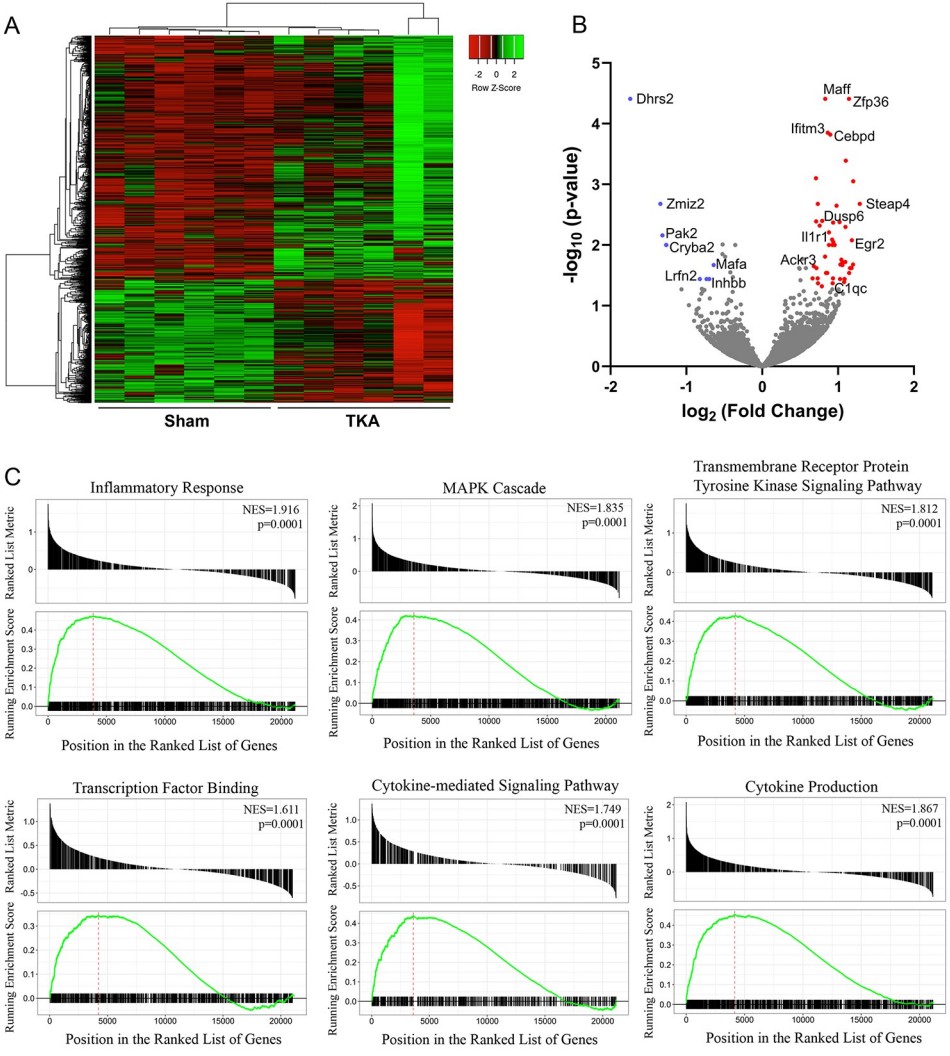

**Fig 7. DRG transcriptome evaluation 24h after TKA surgery. (A)** Hierarchically clustered heat map comparing DRG transcriptomes between individual Sham and TKA rats (n = 6). (**B**) Volcano plot showing relative expression of DRG transcriptomes from TKA rats compared to Sham. The x-axis shows the log2 fold change and the y-axis shows the -$\log_{10}$ adjusted p values. Thresholds for significant changes in expression were set at ≥1.5 fold-change and adjusted p values <0.05. Downregulated and upregulated genes are shown in blue and red, respectively. Several upregulated genes of interest are labeled. (**C**) Gene Set Enrichment Analysis (GSEA) plots for upregulated pathways in DRGs from TKA rats compared to Sham.

we filtered the data by focusing on genes exhibiting ≥1.5 fold-change, resulting in 56 transcripts (Fig 7B and Table 1). Gene Set Enrichment Analysis (GSEA) and pathway analyses for these transcripts identified highly ranked pathways associated with inflammatory response (*Il1r1*, *Apod*, *Zfp36*, *Per1*, *Tnfaip6*, *Socs3*, *Serpine1*), cytokine production (*Il1r1*, *Apod*, *Per1*, *Agpat2*), MAPK (*Zfp36*, *Dusp6*, *Map3k6*, *Ackr3*, *P2ry6*, *Per1*, *Myc*), transcription factor binding (*Zfp36*, *Egr2*, *Per1*, *Dcn*, *Myc*, *Socs3*), cytokine-mediated signaling (*Ifitm3*, *Ackr3*), and transmembrane receptor protein tyrosine kinase signaling (*Apod*, *Per1*, *Myc*, *Socs3*) (Fig 7C). Next, we queried a DRG single cell transcriptomic database to identify the potential cellular origin of these transcripts and found that the majority mapped to macrophages (*Zfp36*, *Maff*, *Egr2*, *Fosl2*, *P2ry6*, *Socs3*, *Serpine1*) and/or Schwann cells (*Cebpd*, *Cldn1*, *Steap4*, *Apod*, *Egr2*, *Ackr3*, *Tnfaip6*, *Socs3*) [36]. We also compared sham to the naïve group and identified *Ccl2* as the sole upregulated transcript (S1 Table).

## Discussion

TKA is the final option to decrease pain and improve mobility for patients with end-stage OA. However, TKA surgery induces acute postoperative pain that is higher than for other joints such at the hip [37]. The development of new pain management strategies for TKA patients necessitates an understanding of the mechanisms underlying acute post-TKA pain. Toward this end, we developed a rat TKA model that displays acute postoperative pain and assessed the corresponding DRG transcriptomic signatures.

TKA is an invasive procedure that requires removal of femoral and tibial condyles (cartilage and significant portions of the subchondral bone), intramedullary reaming, and implantation of metallic prosthesis (press-fit or cemented), as well as tensioning and reconstruction of the joint capsule. While several rat models of knee arthroplasty have been reported [14–16], as well as a few less invasive models of knee surgery [29,31], none of these models have been shown to replicate the acute postoperative pain and restoration of function as observed clinically [38–40]. Our TKA model mimics the clinical procedure: trimming of the tibial and femoral condyles, intramedullary drilling and reaming, implantation of tibial and femoral prostheses, as well as reconstruction of the knee joint. Radiographic imaging confirmed proper positioning of tibial and formal prostheses in full extension and flexion and restoration of normal range of motion within 35 days (Fig 2I and 2J, S4 Video).

The postoperative period in TKA patients is characterized by an acute inflammatory phase and acute pain at rest and upon ambulation that gradually recovers. Our model reflects these clinical observations as TKA was accompanied by elevated IL-6 and TNFα levels and functional deficits consistent with acute postoperative pain. Morris et al. reported a similar pattern of elevated IL-6, though not to a significant degree, following knee surgery in rats [17]. The animals were observed initially limping and then gradually recovered normal gait, confirming that rats subjected to TKA regained normal knee loading and function. We employed multiple behavioral outcomes to assess postoperative pain and function, with each exhibiting distinct recovery kinetics. The rate of functional recovery for this TKA model is slower compared to less severe models of knee surgery [29,31], which typically exhibit functional recoveries within a week. However, complete functional recovery is demonstrated within five weeks, in contrast to a recent model that exhibited continued functional deficits at 3 months [14]. We developed our model in healthy rats and subsequently ascertained any potential postoperative differences in rats with preexisting OA. This comparison revealed that the magnitude and recovery kinetics were largely similar between the two groups, with the sole exceptions of a greater loss of SFI in the OA group at day 7 and prolonged deficits in stride length until day 14.

**Table 1. Differentially expressed transcripts in DRGs 24h after surgery (TKA/sham).**

| Ensembl ID | Gene name | log2FC | Adjusted p value | Name |
|---|---|---|---|---|
| ENSRNOT00000090129 | Zfp36 | 1.142 | 3.92E-05 | ZFP35 ring finger protein |
| ENSRNOT00000017181 | Maff | 0.829 | 3.92E-05 | V-maf avian musculoaponeurotic fibrosarcoma oncogene homolog F |
| ENSRNOT00000024483 | Dhrs2 | -1.742 | 3.92E-05 | Dehydrogenase/reductase (SDR family) member 2 |
| ENSRNOT00000020265 | Ifitm3 | 0.863 | 0.000143 | Interferon induced transmembrane protein 3 |
| ENSRNOT00000074586 | Cebpd | 0.896 | 0.000150 | CCAAT/enhancer binding protein (C/EBP), delta |
| ENSRNOT00000002640 | Cldn1 | 1.101 | 0.000404 | Claudin 1 |
| ENSRNOT00000061041 | Rpl26-ps2 | 0.708 | 0.000788 | Ribosomal protein L26 |
| ENSRNOT00000010289 | Apold1 | 1.198 | 0.000888 | Apolipoprotein L domain containing 1 |
| ENSRNOT00000011432 | Steap4 | 1.285 | 0.002094 | STEAP family member 4 |
| ENSRNOT00000052002 | Zmiz2 | -1.343 | 0.002094 | Zinc finger, MIZ-type containing 2 |
| ENSRNOT00000022144 | Enc1 | 0.732 | 0.002105 | Ectodermal-neural cortex 1 (with BTB domain) |
| ENSRNOT00000007320 | Best1 | 0.979 | 0.002238 | Bestrophin 1 |
| ENSRNOT00000037844 | Dusp6 | 0.790 | 0.004010 | Dual specificity phosphatase 6 |
| ENSRNOT00000083188 | Clec2g | 0.710 | 0.004118 | C-type lectin domain family 2, member g |
| ENSRNOT00000047635 | Elf2 | 1.020 | 0.004179 | E74-like factor 2 (ets domain transcription factor) |
| ENSRNOT00000009894 | Nfkbia | 0.934 | 0.004237 | Nuclear factor of kappa light polypeptide gene enhancer in B-cells inhibitor, alpha |
| ENSRNOT00000024947 | Tubb6 | 0.757 | 0.004794 | Tubulin, beta 6 class V |
| ENSRNOT00000073330 | Apod | 1.097 | 0.005026 | Apolipoprotein D |
| ENSRNOT00000082146 | Il1r1 | 0.878 | 0.006134 | Interleukin 1 receptor, type I |
| ENSRNOT00000071060 | Pak2 | -1.317 | 0.006965 | P21 protein (Cdc42/Rac)-activated kinase 2 |
| ENSRNOT00000056414 | Col6a6 | 0.922 | 0.008089 | Collagen, type VI, alpha 6 |
| ENSRNOT00000000792 | Egr2 | 1.181 | 0.008351 | Early growth response 2 |
| ENSRNOT00000024011 | Klf15 | 0.938 | 0.008817 | Kruppel-like factor 15 |
| ENSRNOT00000026186 | Sult1a1 | 0.879 | 0.009924 | Sulfotransferase family, cytosolic, 1A, phenol-preferring, member 1 |
| ENSRNOT00000077894 | Fosl2 | 0.954 | 0.009924 | FOS-like antigen 2 |
| ENSRNOT00000021403 | Nabp1 | 0.925 | 0.009924 | Nucleic acid binding protein 1 |
| ENSRNOT00000024283 | Cryba2 | -1.268 | 0.009924 | Crystallin, beta A2 |
| ENSRNOT00000005797 | Dlx3 | -1.265 | 0.010100 | Distal-less homeobox 3 |
| ENSRNOT00000012463 | Map3k6 | 0.828 | 0.015538 | Mitogen-activated protein kinase kinase kinase 6 |
| ENSRNOT00000025856 | Ch25h | 1.042 | 0.017358 | Cholesterol 25-hydroxylase |
| ENSRNOT00000074993 | Lrg1 | 1.094 | 0.019213 | Leucine-rich alpha-2-glycoprotein 1 |
| ENSRNOT00000063831 | Zswim8 | 1.056 | 0.019213 | Zinc finger, SWIM-type containing 8 |
| ENSRNOT00000035123 | Tmem252 | 1.196 | 0.020854 | Transmembrane protein 252 |
| ENSRNOT00000000628 | Cdkn1a | 1.030 | 0.020854 | Cyclin-dependent kinase inhibitor 1A |
| ENSRNOT00000077727 | Pla1a | 1.058 | 0.021155 | Phospholipase A1 member A |
| ENSRNOT00000010084 | Mafa | -0.642 | 0.021502 | MAF bZIP transcription factor A |
| ENSRNOT00000026558 | Ackr3 | 0.678 | 0.022403 | Atypical chemokine receptor 3 |
| ENSRNOT00000050227 | P2ry6 | 1.176 | 0.024179 | Pyrimidinergic receptor P2Y6 |
| ENSRNOT00000023488 | Gja5 | 1.156 | 0.024179 | Gap junction protein, alpha 5 |
| ENSRNOT00000057136 | Per1 | 0.714 | 0.024179 | Period circadian regulator 1 |
| ENSRNOT00000070792 | Tnfaip6 | 1.146 | 0.028773 | TNF alpha induced protein 6 |
| ENSRNOT00000006070 | Dcn | 0.857 | 0.028773 | Decorin |
| ENSRNOT00000031701 | Rnf122 | 0.842 | 0.028773 | Ring finger protein 122 |
| ENSRNOT00000021353 | Lrp4 | 0.729 | 0.035766 | LDL receptor related protein 4 |
| ENSRNOT00000014388 | Col8a2 | 0.663 | 0.035766 | Collagen type VIII alpha 2 chain |
| ENSRNOT00000006188 | Myc | 0.932 | 0.035766 | MYC proto-oncogene, bHLH transcription factor |
| ENSRNOT00000015705 | Lrfn2 | -0.823 | 0.035906 | Leucine rich repeat and fibronectin type III domain containing 2 |

*(Continued)*

**Table 1.** (Continued)

| Ensembl ID | Gene name | log2FC | Adjusted p value | Name |
|---|---|---|---|---|
| ENSRNOT00000003940 | Socs3 | 1.025 | 0.035906 | Suppressor of cytokine signaling 3 |
| ENSRNOT00000001916 | Serpine1 | 1.086 | 0.035906 | Serpin family E member 1 |
| ENSRNOT00000086350 | Inhbb | -0.698 | 0.035906 | Inhibin subunit beta B |
| ENSRNOT00000013284 | Piwil2 | -0.735 | 0.035906 | Piwi-like RNA-mediated gene silencing 2 |
| ENSRNOT00000067391 | Mt2A | 1.084 | 0.036556 | Metallothionein 2A |
| ENSRNOT00000073595 | LOC688459 | 1.078 | 0.041160 | Hypothetical protein LOC688459 |
| ENSRNOT00000017065 | C1qc | 0.928 | 0.042207 | Complement C1q C chain |
| ENSRNOT00000026408 | Agpat2 | 0.735 | 0.042458 | 1-acylglycerol-3-phosphate O-acyltransferase 2 |
| ENSRNOT00000029431 | Cep76 | 0.786 | 0.047454 | Centrosomal protein 76 |

Following TKA surgery, the rats responded to clinically utilized perioperative analgesics, although it is notable that neither morphine nor ketorolac fully restored incapacitance to baseline levels when used individually or when co-administered. These results are consistent with clinical reports of acute post-TKA pain, which persists despite the use of multimodal analgesia [7–10]. Interestingly, one of the previously described knee surgery models was characterized by full reversal of pain-associated behaviors using the same analgesics [29]. However, as this model does not involve the placement of implants, it likely induces less severe acute postoperative pain compared to TKA surgery [7–10]. Our results for gabapentin were partially consistent with clinical findings reporting limited or no efficacy of gabapentin in treating post-TKA pain [34,35]. Specifically, gabapentin improved SFI while it failed to alter incapacitance, suggesting that incapacitance may be a more appropriate surrogate for acute post-TKA pain in this model.

Subsequently, we leveraged our model to identify differentially regulated molecular pathways in DRGs after TKA, selecting a 24h time point to reflect acute postoperative pain. Transcripts associated with inflammation, cytokine signaling, MAPK pathway, and transcription factors were enriched in our analysis. Utilizing established single-cell DRG RNA-seq datasets, it is likely that these transcripts originate from macrophages and Schwann cells, both of which release cytokines and chemokines and are implicated in the pathogenesis of pain [41–43]. This is consistent with previous work demonstrating that depletion of myeloid cells (including macrophages) alleviates postoperative pain hypersensitivity [43]. A subset of differentially regulated transcripts observed in our study (e.g., C1qc) are likewise altered in other pain models [44]. Interestingly, the endogenous opioid scavenger Ackr3 [45] was upregulated in TKA rats, and it is tempting to speculate that this may reflect reduced endogenous opioid tone leading to enhanced pain sensitivity.

A recent study of the DRG proteome identified 44 proteins differentially regulated following plantar incision in mice [46], with Annexin A1 emerging as a potential mediator of pain. Surprisingly, there was no overlap between the post-incisional proteomic profile and our transcriptomic results, suggesting that TKA and plantar incision induce distinct changes in gene expression. Furthermore, it should be noted that our sham condition involved anesthesia, skin incision, and suturing, while the proteomic study employed anesthesia alone. Indeed, when comparing the transcriptome of TKA rats to naive controls, 342 transcripts were differentially regulated, including Annexin A1 (S1 Table). Collectively, our results identify transcripts and molecular pathways that are upregulated in response to TKA surgery and highlight the potential roles for macrophages and Schwann cells in acute postsurgical pain.

The ability of this model to replicate TKA surgery and underlying OA pathology while controlling variables such as age, sex, weight, and disease status, may make it a valuable tool to

investigate clinical questions regarding post-TKA pain and functional outcomes. This model can also serve as a platform to evaluate novel analgesics and interventions to enhance recovery after surgery. One limitation of the current study is the use of rats as a model system, a quadruped with differing gait patterns compared to humans. Additionally, as the rats preferentially maintained their hind limbs that underwent TKA in an elevated position for several days after surgery, it was not possible to assess evoked pain responses such as paw withdrawal in response to mechanical or thermal stimuli.

## Conclusions

We developed a rat TKA model that closely replicates the intraoperative procedures performed during clinical TKA surgery. The model results in postoperative changes in behavior and gait that return to baseline within five weeks. We further show that it can be used to assess postoperative behaviors indicative of pain to evaluate analgesic efficacy. Lastly, our model provides access to postoperative tissues, such as DRGs, to elucidate how changes in gene expression may impact postoperative pain and recovery.

## Supporting information

**S1 Table. Comparison of DRG transcript levels between TKA and Sham, TKA and naïve, and Sham and naïve groups 24h after TKA or Sham surgery.**
(XLSX)

**S1 Video. TKA positioning and range of motion.** Video fluoroscopy showing the positioning of the femoral and tibial prostheses through the full range of motion immediately post-implantation.
(MP4)

**S2 Video. Normal gait at baseline.**
(MP4)

**S3 Video. Gait on day 2 after TKA.**
(MP4)

**S4 Video. Gait on day 35 after TKA.**
(MP4)

## Author Contributions

**Conceptualization:** David E. Komatsu, Martin Kaczocha.

**Data curation:** David E. Komatsu, Chris Gordon, Martin Kaczocha.

**Formal analysis:** David E. Komatsu, Sardar M. Z. Uddin, Jason Shieh, Agnieszka B. Bialkowska, Martin Kaczocha.

**Funding acquisition:** Martin Kaczocha.

**Investigation:** David E. Komatsu, Sardar M. Z. Uddin, Chris Gordon, Martha P. Kanjiya, Diane Bogdan, Justice Achonu, Adriana DiBua, Hira Iftikhar, Amanda Ackermann, Rohan J. Shah, Martin Kaczocha.

**Methodology:** David E. Komatsu, Sardar M. Z. Uddin, Martin Kaczocha.

**Project administration:** David E. Komatsu, Martin Kaczocha.

**Supervision:** David E. Komatsu, Martin Kaczocha.

**Visualization:** Jason Shieh, Agnieszka B. Bialkowska, Martin Kaczocha.

**Writing – original draft:** David E. Komatsu, Sardar M. Z. Uddin, Martin Kaczocha.

**Writing – review & editing:** David E. Komatsu, Sardar M. Z. Uddin, Martin Kaczocha.

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
