## [Decision Letter · Decision Letter 0]

26 Sep 2022

PONE-D-22-16993Acute postoperative pain and dorsal root ganglia transcriptomic signatures following total knee arthroplasty in ratsPLOS ONE

Dear Dr. 

Thank you for submitting your manuscript to PLOS ONE. After careful consideration, we feel that it has merit but does not fully meet PLOS ONE’s publication criteria as it currently stands. Therefore, we invite you to submit a revised version of the manuscript that addresses the points raised during the review process.

We look forward to receiving your revised manuscript.

Kind regards,

Rosanna Di Paola, MD

Academic Editor

PLOS ONE

Journal Requirements:

3. As part of your revision, please complete and submit a copy of the Full ARRIVE 2.0 Guidelines checklist, a document that aims to improve experimental reporting and reproducibility of animal studies for purposes of post-publication data analysis and reproducibility: https://arriveguidelines.org/sites/arrive/files/Author%20Checklist%20-%20Full.pdf (PDF). Please include your completed checklist as a Supporting Information file. Note that if your paper is accepted for publication, this checklist will be published as part of your article.

Reviewers' comments:

Reviewer's Responses to Questions

**Comments to the Author**

1. Is the manuscript technically sound, and do the data support the conclusions?

Reviewer #1: Partly

Reviewer #2: No

2. Has the statistical analysis been performed appropriately and rigorously? 

Reviewer #1: Yes

Reviewer #2: N/A

3. Have the authors made all data underlying the findings in their manuscript fully available?

Reviewer #1: Yes

Reviewer #2: Yes

4. Is the manuscript presented in an intelligible fashion and written in standard English?

Reviewer #1: Yes

Reviewer #2: Yes

5. Review Comments to the Author

Reviewer #1: The study was performed at a high level, but there are several technical issues that require correction for publication.

To make a decision on further publication, it is necessary to revise the indicated sections and add mandatory papers with a second review:

+ A shorter and more capacious name that meets modern trends: "Acute postoperative pain and dorsal root ganglia transcriptomic signatures following TKA in rats: experimental study"

+ In the introduction and relevance, it is better not to refer to papers that are 15 years old, it is better to find references up to 5 years old. There are also recent studies regarding the level of pain after TKA, for example DOI: 10.1007/s00167-017-4713-5

+ Where is the conclusion section? The results must be summarized.

+ Line 474 – What document is the link to? And then the URL: ...

+ A level of evidence needs to be added.

Reviewer #2: As the authors point out, there is a need for better animal models of human health problems. Fig 1-3 concern the surgery and functional recovery profile of a rat model of knee replacement. Fig 6 reports RNA-seq data identifying upregulated transcripts and pathways in the acute postoperative period.

My main concern is that the appropriate controls are missing from the figures seeking to demonstrate that clinically utilized perioperative analgesics can reduce the behavioral measures of recovery (Fig 4), and to investigate if the recovery from knee replacement in animals with preexisting OA is altered (Fig 5). The absence of these controls impacts on their validity. A vehicle control is needed in Fig 4 and an OA - sham group needs to be added to Fig 5.

minor points:

I prefer a graph reporting inflammatory mediator changes. Are the levels you measured over the recovery period what you expected based on previously published data?

Fig 4E, why is the before morphine SFI elevated? is this significant cf to before index for the other two groups?

Reconsider how to frame the abstract/introduction and start of discussion. I am not convinced that the severe post-TKA pain state observed in people with OA following TKA is the most compelling argument to use, and the data presented doesn't address the need for larger doses of analgesics at all.

Take care to use the correct terminology when discussing the interpretation of the behavioral outputs measured. Changes in locomotion, rearing and gait are not "pain" per se.

The argument that functional recovery is only partially improved by analgesia would be clearer if incapacitance, SFI, and stride length values from sham animals were somehow indicated in fig 4

Include more information about why DMM was chosen as a OA model and state how long it persists (is the incapacitance level reported in Fig 5C stable for the whole duration of the experimental protocol)

The axis text in Fig 6C is blurred and can not be read in the version supplied. What transcripts were down-regulated in your study?

Would you expect morphine/ketorolac treatment to fully restore incapacitance? In addition to being a pain associated behavior, what else does it capture?

436: .....may reflect reduced endogenous opioid tone leading to enhanced pain sensitivity - but see https://doi.org/10.1016/j.neuron.2021.04.011 and https://doi.org/10.1016/j.neuron.2021.03.012

6. PLOS authors have the option to publish the peer review history of their article (what does this mean?). If published, this will include your full peer review and any attached files.

Reviewer #1: No

Reviewer #2: No

---

## [Author Response · Author response to Decision Letter 0]

18 Oct 2022

¬¬¬Reviewer 1

A shorter and more capacious name that meets modern trends: "Acute postoperative pain and dorsal root ganglia transcriptomic signatures following TKA in rats: experimental study"

 We have changed the title in accordance with the reviewer’s suggestion.

In the introduction and relevance, it is better not to refer to papers that are 15 years old, it is better to find references up to 5 years old. There are also recent studies regarding the level of pain after TKA, for example DOI: 10.1007/s00167-017-4713-5

The older citations have been removed and newer citations added, including the suggested study on post TKA pain.

Where is the conclusion section? The results must be summarized.

 A conclusion section has been added after the discussion. 

Line 474 – What document is the link to? And then the URL: ...

The URL was to a Centers for Disease Control website. This has been removed and a similar manuscript citation has been added.

A level of evidence needs to be added.

 The level of evidence is II and this has been added to the end of the abstract. 

Reviewer 2 

My main concern is that the appropriate controls are missing from the figures seeking to demonstrate that clinically utilized perioperative analgesics can reduce the behavioral measures of recovery (Fig 4), and to investigate if the recovery from knee replacement in animals with preexisting OA is altered (Fig 5). The absence of these controls impacts on their validity. A vehicle control is needed in Fig 4 and an OA - sham group needs to be added to Fig 5.

The reviewer makes a valid point; however, these analyses were designed as a within group pre- to post-drug response and not as a between group vehicle control study. Nevertheless, we did collect vehicle data for incapacitance, which expectedly showed no effect. These data have been added to the revised figure 5 (previously figure 4). Unfortunately, vehicle controls were not collected on the SFI and stride length analyses, so these could not be added. However, saline was used as a vehicle for all of the drugs employed and thus would not be expected to alter SFI or stride length.

As for the OA-sham in Figure 6 (previously figure 5), the intent of this series of experiments was to determine if preexisting OA influenced postoperative pain and recovery following TKA surgery by performing a within group comparison to presurgical baseline values. We elected to use a sham+sham control to determine if repeated surgical procedures affected our outcomes, and it did not. We did not actually compare sham+sham to OA+sham. Moreover, this study was not designed to compare TKA surgery to the natural progression of OA to see if surgery alleviates long-term pain and function. However, this is an intriguing idea that we may explore in a subsequent study. 

Minor points:

I prefer a graph reporting inflammatory mediator changes. Are the levels you measured over the recovery period what you expected based on previously published data?

Fig 4E, why is the before morphine SFI elevated? is this significant cf to before index for the other two groups?

 We have changed table 1 to the new figure 4. 

As for the apparent before morphine elevation, we believe that this merely represents variability observed in this model. This also supports our decision to use a within group pre-post design so that such differences do not adversely impact our results. We did compare the before levels between all groups and they did not significantly differ. 

We identified a reference to IL-6 elevation similar to what we observed for a slightly different rat knee surgical model and have added that to the discussion. TNFa is elevated in many OA models, but we could find no references to it after surgical procedures in rats. 

Reconsider how to frame the abstract/introduction and start of discussion. I am not convinced that the severe post-TKA pain state observed in people with OA following TKA is the most compelling argument to use, and the data presented doesn't address the need for larger doses of analgesics at all.

We have made minor modifications to these sections; however clinical data support the assertion that pain is higher in TKA patient than other surgeries and this is one of the reasons for the development of this model. We have removed the reference to the clinical use of larger doses of analgesics. 

Take care to use the correct terminology when discussing the interpretation of the behavioral outputs measured. Changes in locomotion, rearing and gait are not "pain" per se.

 We agree that these are surrogates of pain and have modified our manuscript accordingly. 

The argument that functional recovery is only partially improved by analgesia would be clearer if incapacitance, SFI, and stride length values from sham animals were somehow indicated in fig 4

The graphs in figure 5 (previously fig 4) now have lines indicating mean sham values added to them. 

Include more information about why DMM was chosen as a OA model and state how long it persists (is the incapacitance level reported in Fig 5C stable for the whole duration of the experimental protocol)

 Justification for the DMM model has been added to the Methods section under DMM surgery and is supported by a new reference. We did not continue to measure incapacitance levels beyond 4 weeks for this experiment. However, this model has been studied out to 12 weeks and shows no further changes in incapacitance after 4 weeks (PMID: 27856294, 34136155). 

The axis text in Fig 6C is blurred and can not be read in the version supplied. What transcripts were down-regulated in your study?

 The font sizes of axes have been increased and now appears clearer. In addition, downregulated transcripts have been added to the volcano plot. These are also listed in Table 1 (formerly Table 2).

Would you expect morphine/ketorolac treatment to fully restore incapacitance? In addition to being a pain associated behavior, what else does it capture?

 Our goal in combining morphine/ketorolac was to model commonly used perioperative analgesics. We hypothesized that the combination would be more efficacious than either drug alone but did not expect them to fully restore incapacitance. Incapacitance may also capture differences in proprioception and nerve dysfunction leading to differential weight bearing. 

436: .....may reflect reduced endogenous opioid tone leading to enhanced pain sensitivity - but see https://doi.org/10.1016/j.neuron.2021.04.011 and https://doi.org/10.1016/j.neuron.2021.03.012

 We are indeed aware of this study and agree that in contrast to the enhanced opioid tone observed upon NaV1.7 deletion, we do not know whether endogenous opioids contribute to pain in our model. Nevertheless, our transcriptomic data offer a tantalizing suggestion of altered opioid function and therefore we highlighted this possibility.

---

## [Decision Letter · Decision Letter 1]

14 Nov 2022

PONE-D-22-16993R1Acute postoperative pain and dorsal root ganglia transcriptomic signatures following total knee arthroplasty (TKA) in rats: An experimental studyPLOS ONE

Dear Dr. Kaczocha,

Thank you for submitting your manuscript to PLOS ONE. After careful consideration, we feel that it has merit but does not fully meet PLOS ONE’s publication criteria as it currently stands. Therefore, we invite you to submit a revised version of the manuscript that addresses the points raised during the review process.

Both reviewers have raised minor points. The correction of these points would further improve the manuscript. Please amend the paper according to their suggestions.

We look forward to receiving your revised manuscript.

Kind regards,

Antal Nógrádi, M.D., Ph.D., D.Sc.

Academic Editor

PLOS ONE

Journal Requirements:

Reviewers' comments:

Reviewer's Responses to Questions

**Comments to the Author**

1. If the authors have adequately addressed your comments raised in a previous round of review and you feel that this manuscript is now acceptable for publication, you may indicate that here to bypass the “Comments to the Author” section, enter your conflict of interest statement in the “Confidential to Editor” section, and submit your "Accept" recommendation.

Reviewer #1: All comments have been addressed

Reviewer #2: All comments have been addressed

2. Is the manuscript technically sound, and do the data support the conclusions?

Reviewer #1: Yes

Reviewer #2: Yes

3. Has the statistical analysis been performed appropriately and rigorously? 

Reviewer #1: Yes

Reviewer #2: Yes

4. Have the authors made all data underlying the findings in their manuscript fully available?

Reviewer #1: Yes

Reviewer #2: Yes

5. Is the manuscript presented in an intelligible fashion and written in standard English?

Reviewer #1: Yes

Reviewer #2: Yes

6. Review Comments to the Author

Reviewer #1: Good job on paper! The only thing for the correct presentation in the reference databases and the correct presentation of the work is to categorize the abstract (aim, materials&methods, results, etc.).

Reviewer #2: Thanks to the authors for responding to concerns raised and clarifying some aspects in the text. Fig 6 - especially 6C is still very difficult to read and needs to be improved for publication.

Should Orthop Surg Res. 2021 Dec 20;16(1):726. doi: 10.1186/s13018-021-02871-y. be referenced in the introduction?

7. PLOS authors have the option to publish the peer review history of their article (what does this mean?). If published, this will include your full peer review and any attached files.

Reviewer #1: No

Reviewer #2: No

---

## [Author Response · Author response to Decision Letter 1]

15 Nov 2022

¬¬¬Reviewer 1

Good job on paper! The only thing for the correct presentation in the reference databases and the correct presentation of the work is to categorize the abstract (aim, materials&methods, results, etc.).

 We thank the reviewer for this suggestion. We would like to note that our Abstract is formatted in accordance with journal guidelines (i.e., not categorized).

Reviewer 2 

Thanks to the authors for responding to concerns raised and clarifying some aspects in the text. Fig 6 - especially 6C is still very difficult to read and needs to be improved for publication. Should Orthop Surg Res. 2021 Dec 20;16(1):726. doi: 10.1186/s13018-021-02871-y. be referenced in the introduction?

In accordance with the reviewer’s suggestion, we have enlarged the graphs presented in panel C. 

We have also included the aforementioned reference (now #17) in the introduction.

---

## [Editor Report · Decision Letter 2]

21 Nov 2022

Acute postoperative pain and dorsal root ganglia transcriptomic signatures following total knee arthroplasty (TKA) in rats: An experimental study

PONE-D-22-16993R2

Dear Dr. Kaczocha,

We’re pleased to inform you that your manuscript has been judged scientifically suitable for publication and will be formally accepted for publication once it meets all outstanding technical requirements.

Kind regards,

Antal Nógrádi, M.D., Ph.D., D.Sc.

Academic Editor

PLOS ONE
---

## [Editor Report · Acceptance letter]

23 Nov 2022

PONE-D-22-16993R2 

Acute postoperative pain and dorsal root ganglia transcriptomic signatures following total knee arthroplasty (TKA) in rats: An experimental study 

Dear Dr. Kaczocha:

I'm pleased to inform you that your manuscript has been deemed suitable for publication in PLOS ONE. Congratulations! Your manuscript is now with our production department. 

Kind regards, 

on behalf of

Prof. Antal Nógrádi 

Academic Editor

PLOS ONE